# The CD8+ and CD4+ T Cell Immunogen Atlas of Zika Virus Reveals E, NS1 and NS4 Proteins as the Vaccine Targets

**DOI:** 10.3390/v14112332

**Published:** 2022-10-25

**Authors:** Hangjie Zhang, Wenling Xiao, Min Zhao, Yingze Zhao, Yongli Zhang, Dan Lu, Shuangshuang Lu, Qingxu Zhang, Weiyu Peng, Liumei Shu, Jie Zhang, Sai Liu, Kexin Zong, Pengyan Wang, Beiwei Ye, Shihua Li, Shuguang Tan, Fuping Zhang, Jianfang Zhou, Peipei Liu, Guizhen Wu, Xuancheng Lu, George F. Gao, William J. Liu

**Affiliations:** 1Department of Immunization Program, Zhejiang Provincial Center for Disease Control and Prevention, Hangzhou 310021, China; 2NHC Key Laboratory of Biosafety, National Institute for Viral Disease Control and Prevention, Chinese Center for Disease Control and Prevention (China CDC), Beijing 102206, China; 3Shunde Hospital, Guangzhou Medical University (The Lecong Hospital of Shunde, Foshan), Foshan 528000, China; 4CAS Key Laboratory of Pathogen Microbiology and Immunology, Institute of Microbiology, Chinese Academy of Sciences, Beijing 100101, China; 5Laboratory of Animal Center, Chinese Center for Disease Control and Prevention, Beijing 102206, China

**Keywords:** Zika virus, T cells, immunodominance, peptide, epitope

## Abstract

Zika virus (ZIKV)-specific T cells are activated by different peptides derived from virus structural and nonstructural proteins, and contributed to the viral clearance or protective immunity. Herein, we have depicted the profile of CD8+ and CD4+ T cell immunogenicity of ZIKV proteins in C57BL/6 (H-2^b^) and BALB/c (H-2^d^) mice, and found that featured cellular immunity antigens were variant among different murine alleles. In H-2^b^ mice, the proteins E, NS2, NS3 and NS5 are recognized as immunodominant antigens by CD8+ T cells, while NS4 is dominantly recognized by CD4+ T cells. In contrast, in H-2^d^ mice, NS1 and NS4 are the dominant CD8+ T cell antigen and NS4 as the dominant CD4+ T cell antigen, respectively. Among the synthesized 364 overlapping polypeptides spanning the whole proteome of ZIKV, we mapped 91 and 39 polypeptides which can induce ZIKV-specific T cell responses in H-2^b^ and H-2^d^ mice, respectively. Through the identification of CD8+ T cell epitopes, we found that immunodominant regions E_294-302_ and NS4_2351-2360_ are hotspots epitopes with a distinct immunodominance hierarchy present in H-2^b^ and H-2^d^ mice, respectively. Our data characterized an overall landscape of the immunogenic spectrum of the ZIKV polyprotein, and provide useful insight into the vaccine development.

## 1. Introduction

As a mosquito-borne virus belonging to the flavivirus genus of the *Flaviviridae* family, Zika virus (ZIKV) was firstly isolated in 1947 from rhesus macaque (*Macaca mulatta*) in the Zika forest, Uganda [1,2]. Although human infection was reported as early as 1964, the first major ZIKV outbreak did not occur until 2007 in Yap Island, where over 70% of the population within the island became infected [3]. Infection with ZIKV in humans is often asymptomatic or mild, consisting of skin rashes, conjunctivitis, fever and headaches [4]. However, the outbreak of Zika virus, emerging since 2016 in French Polynesia and in South America and spreading immediately globally, was linked to Guillain–Barre syndrome in adults as well as an increase in fetal abnormalities, including placental insufficiency, microcephaly, making ZIKV infection a global health crisis by the World Health Organization [5,6,7,8,9,10]. Additionally, ZIKV can be transmitted by sexual, blood-borne and maternal-fetal routes [11,12,13], and male infertility has been reported in mouse and human studies [14,15,16].

Studies from mouse models and exposed humans have demonstrated a strong adoptive virus-specific T cells response in clearance of ZIKV [17,18,19,20]. CD4+ T cells proliferate rapidly and have been shown to have an essential role in protection against primary ZIKV infection through assisting B cells to generate neutralizing antibodies and producing polyfunctional cytokines in a murine model [17,21,22,23]. Concomitantly, CD8+ T cells eliminate ZIKV infection by recognizing conserved viral proteins presented by major histocompatibility complex (MHC) class I glycoproteins [24,25], becoming activated and expressing antiviral cytokines, suggesting a protective cytotoxic T-cell response [26,27,28]. Moreover, the depletion of CD4+ and CD8+ T cells or deficiency of T cells in Rag1^−/−^ mice resulted in higher viral loads after infection of ZIKV, but adoptive transfer of CD8+ T cells from ZIKV-infected mice reversed this effect [18,27,28], thus, indicating a pivotal role of T cells in the anti-ZIKV immunity. However, the immunodominant hierarchy of the ZIKV polyprotein is still largely unknown.

The ZIKV genome contains a single open reading frame encoding a polyprotein consisting of 3410 amino acids, which would be post-translationally processed into structural (C, prM/M, and E) and non-structural (NS1, NS2a, NS2b, NS3, NS4a, NS4b, and NS5) proteins by cellular and viral proteases [29]. The antigenic characteristics of the different ZIKV proteins are not well determined. CD4+ and CD8+ T-cell responses to capsid, envelope proteins and non-structural protein 1 (NS1) have been observed in ZIKV-infected monkeys and humans [30,31]. In mice, several CD8+ T-cell epitopes restricted to H-2^b^ have been identified, with a significant portion derived from envelope proteins, including E_294-302_ [27,28]. Moreover, Wen et al. identified HLA-B*0702 and HLA-A*0101-restricted epitopes in *Ifnar1*^−/−^ HLA transgenic mice after ZIKV infection [32]. However, the profile of antigenic peptides spanning the whole ZIKV proteome has not been defined. 

In this study, we characterized the immunogenic hierarchy of ZIKV based on the peptides synthesized spanning all the structural and nonstructural proteins. The profile of immunodominant antigens and epitopes was mapped among the H-2^b^ and H-2^d^ mice. The CD8+ and CD4+ T cell recognition features of the epitope spectrum were characterized. These findings suggest a clear cell-mediated antigenic profile with epitope hotspots among the whole proteome of ZIKV and have important implications for designing vaccines and evaluating T-cell assays.

## 2. Materials and Methods

### 2.1. Viral Strains and Mice

ZIKV strain ZIKA-SMGC-1 (GenBank accession number: KX266255) [15] was amplified in C6/36 mosquito cells and harvested from cell supernatants 7–10 days after infection. Virus was titrated using baby hamster kidney (BHK)-21 cell-based focus-forming units (FFUs). Specific-pathogen-free wild-type mice C57BL/6 (H-2^b^) and BALB/c (H-2^d^) were purchased (Vital River Co., Ltd. Beijing, China), and bred at Laboratory Animal Center, Chinese Center for Disease Control and Prevention. All of the animals were housed in groups of three to five animals in Eurotype II long clear-transparent plastic cages with autoclaved dust-free sawdust bedding. They were fed a pelleted and extruded mouse diet ad libitum and had unrestricted access to drinking water. The light/dark cycle in the room consisted of 12/12 h with artificial light. All experiments were performed following institutional Animal Care and Use Committee-approved animal protocols. C57BL/6 and BALB/c female mice between 6 and 8 weeks of age were intraperitoneally inoculated (i.p.) with 10^4^ focus forming units (FFUs) of ZIKV in a 200 μL volume of 10% FBS/PBS buffer. 

### 2.2. Splenocyte Isolation

Isolation of splenocytes was performed as described previously [33]. ZIKV-infected mice were killed 14 days after infection. The spleens were perfused with PBS immediately and disrupted and passed through a 40 µm sieve mechanically. Red blood cells were lysed with RBC lysis solution (Solarbio, Beijing, China) before cryopreservation. Splenocytes were isolated and used for ELISPOT assays and intracellular cytokine staining (ICS) assays.

### 2.3. Peptide Prediction Approaches and Peptide Synthesis

ZIKV polyprotein sequences of Asian lineages (Brazil 2015 strain, GenBank: KU365777) were obtained from the NCBI protein database. Peptides (20-mer) that overlapped by 10 amino acids were designed using online software (www.hiv.lanl.gov, accessed on 12 December 2016) [34]. A total of 364 overlapping polypeptides were designed and synthesized. The 8- to 12-mer epitopes that bound H-2^b^ and H-2^d^ were predicted within the 20-mer peptides using the NetMHC 4.0 Server (http://www.cbs.dtu.dk, accessed on 4 July 2017), as previously described [35]. For each mouse allele, the lists of peptides obtained above were sorted by predicted affinity and restricted to the top 1~3. Overlapping 20-mer peptides and 8- to 12-mer epitope candidates were synthesized by Scilight Biotechnology Co., Ltd. (Beijing, China). The purity of the synthesized peptides was 95%, as determined by high-performance liquid chromatography. Peptides were dissolved in DMSO at 20 mg/mL and stored at −20 °C.

### 2.4. ELISPOT Assays 

Positive overlapping peptides of the ZIKV polyprotein were detected by 2-D matrix pool analysis and further verified with individual peptides. The 364 overlapping peptides were coded and mixed in 80 matrix peptide pools (X-axis:1–1 to 4–12, Y-axis:1–A to 4–G) Appendix A and detected using an IFN-γ ELISPOT assay (BD Pharmingen, San Diego, CA, USA) [34,36]; the positive peptides were detected and verified additionally. Briefly, a total of 5 × 10^5^ mouse splenocytes was stimulated with matrix peptide pools (with 2 μM of each peptide) or 10 µM of individual peptide in 96-well flat-bottom plates that were coated with anti-IFN-γ mAb. Phorbol-12-myristate-13-acetate (PMA) and ionomycin were used as a positive control, whereas DMSO with the mean concentration in peptide/splenocytes co-incubation well was added into the control well (splenocytes alone). After incubation for 20 h, biotinylated IFN-γ mAb was added, followed by streptavidin-HRP. Then 3-amino-9-ethylcarbazole substrate solution was added to the wells and incubated for 5 to 20 min in the dark at room temperature. Finally, IFN-γ spot-forming cells (SFCs) were counted using an ELISPOT reader. Responses are expressed as number of SFCs per 1 × 10^6^ splenocytes and were considered positive if the magnitude of the response was SFCs > 40, the magnitude of the positive well should have 2-folds than the control well. 

### 2.5. Flow Cytometry Analyses

Intracellular cytokine staining assays were conducted as previously described [27,37]. Briefly, splenocytes (2.5 × 10^6^ per sample) were cultured in 10% FBS/RPMI medium supplemented with ZIKV protein peptide pools with 2 μM of each peptide or 10 µM of individual peptide for 4 h at 37 °C in 96-well U-bottom plates. Splenocytes stimulated with PMA-ionomycin were used as a positive control, whereas DMSO with the mean concentration in peptide/splenocytes co-incubation well was added into the control well (splenocytes alone). Brefeldin A (GolgiPlug, BD Biosciences) was then added and incubated with the cells for 2 h before staining. The cells were next incubated for 30 min at 4 °C with PE-conjugated anti-CD3 mAb (Clone 17A2), PerCP-Cy5.5-conjugated anti-CD8 mAb (Clone 53-5.8) and PE-Cy7-conjugated anti-CD4 mAb (clone GK1.5). Subsequently, the cells were permeabilized in Cytofix/Cytoperm for 20 min at 4 °C, washed three times with Perm/Wash buffer, and incubated in the same buffer for 30 min at 4 °C with FITC-conjugated anti-IFN-γ mAb (clone XMG1.2), APC-conjugated anti-IL-2 mAb (clone JES6-5H4), and PE-Cy7-conjugated anti-TNF-α mAb (clone MP6-XT22).

### 2.6. Tetramer Preparation and Staining

H-2^b^-restricted tetramers of peptides E_294–302_, E_345–355_, NS1_1237–1245_, NS2_1479–1486_, NS3_1759–1767_, NS4_2140–2147_ and NS5_2839–2848_ were prepared as previously described [33]. Briefly, to produce biotinylated peptide-MHC protein, H-2D^b^-heavy chain with a specific biotinylation site was modified at the C terminus of the α3 domain. The soluble H-2D^b^/peptide complex was generated through recombinant H-2D^b^ and β_2_m refolded in the presence of high concentrations of H-2D^b^-restricted peptide. Then the H-2D^b^/peptide complexes were purified over a Superdex 200HR column (GE Healthcare) and biotinylated by incubation with D-biotin, ATP, and the biotin protein ligase BirA (Avidity) at 4 °C overnight. The biotinylated H-2D^b^ was further purified over a Superdex 200 10/300 GL gel filtration column (GE Healthcare) to remove excess biotin and then mixed with PE-streptavidin (Sigma-Aldrich). For tetramer and surface marker staining, mouse splenocytes and single-cell suspensions of brain, spinal cord and testicular tissues were incubated with FITC-conjugated anti-CD3 mAb (Clone 17A2), PerCP-Cy5.5-conjugated anti-CD8 mAb (Clone 53-5.8), PE-Cy7-conjugated anti-CD4 mAb (clone GK1.5), and PE-conjugated tetramer at 4 °C in the dark. Multiparameter analyses were performed on a FACSAria™ II (BD Biosciences) and analyzed using FlowJo software (Tree Star).

### 2.7. Statistical Analysis 

Data are expressed as the mean ± SEM. For all analyses, *p*-values were analyzed with Student’s *t*-test (n.s. *p* > 0.05; * *p* < 0.05; ** *p* < 0.01; *** *p* < 0.001). All graphs were analyzed with Prism software version 8.0 (GraphPad Software, Inc. San Diego, CA, USA).

## 3. Results

### 3.1. The Distinct Immunogenic Hierarchy of Structural and Nonstructural Proteins of ZIKV in H-2^b^ and H-2^d^ Mice

To identify the specific peptides and epitopes of ZIKV in C57BL/6 (H-2^b^) and BALB/c (H-2^d^) mice, we designed 364 overlapping peptides from the full-length sequence (3423 amino acids) of ZIKV. Peptides (20-mer) that overlapped by 10 amino acids were synthesized to ensure that shorter peptides (e.g., 8 to 11-mers) were represented in at least one peptide (Figure 1A,B). Next, we tested T-cell responses to the ZIKV protein libraries mixed with peptides from proteins using IFN-γ-ELISPOT assays in H-2^b^ and H-2^d^ mice infected with ZIKV for 14 days (Figure 1C). Robust T-cell reactions can be observed in H-2^b^ mice against E, NS2, NS3, NS4 and NS5 protein libraries, while NS1, NS3, and NS4 protein libraries can induce strong T-cell reactions in H-2^d^ mice (Figure 1D). 

To further validate the profile of the immune reaction to these ZIKV-derived protein libraries, intracellular cytokine staining (ICS) was performed. Splenocytes were stimulated with all eight protein libraries and the frequency of IFN-γ/TNF-α/IL-2-producing CD8+ and CD4+ T cells was determined. E, NS2, NS3 and NS5 protein libraries induced a high frequency of IFN-γ-expressing CD8+ T cells, while, E and NS4 induced a high frequency of IFN-γ-CD4+ T cells in H-2^b^ mice (Figure 2A). In H-2^d^ mice, NS1, NS4 protein libraries induced the highest expression of three cytokines (IFN-γ, IL-2 and TNF-α) in CD8+ T cells, which was similar to the ELISPOT assay results, while NS4 induced the highest IFN-γ-expressing CD4^+^ T cells (Figure 2A,B). Thus, generally, ZIKV E protein in H-2^b^ mice, and NS1 and NS4 in H-2^d^ mice were the dominant antigens for inducing a high frequency of IFN-γ-expressing CD8+ T cells, while NS4 for both mouse alleles dominate the IFN-γ-expressing CD4+ T cell responses. These results demonstrated distinct dominance features of ZIKV protein libraries to induce virus-specific CD8+/CD4+ T cells among different mouse alleles.

### 3.2. The Profile Mapping of Antigenic Peptides across the Whole ZIKV Polyprotein in Mice

To verify the map of the T-cell response to ZIKV, all 364 peptides spanning the ZIKV proteome were tested by IFN-γ-ELISPOT assays using matrix peptide pools in ZIKV-infected wild-type mice. The T-cell responses to ZIKV in H-2^b^ and H-2^d^ mice were not identical, with more H-2^b^-positive epitopes than H-2^d^-restricted ones. Among the eight ZIKV proteins, 91 peptides were positive for H-2^b^ and 39 for H-2^d^. For H-2^b^ mice, positive epitopes were derived from C (1/13), prM (3/17), E (25/53), NS1 (14/38), NS2 (8/37), NS3 (8/66), NS4 (12/41) and NS5 (20/99), with immune hotspots in E and NS1 proteins. For H-2^d^ mice, distribution of the positive peptides among the eight proteins were C (2/13), prM (0/17), E (11/53), NS1 (4/38), NS2 (7/37), NS3 (6/66), NS4 (7/41) and NS5 (2/99) (Figure 3A,B). The frequencies of peptide-specific IFN-γ-producing T cells ranged from 40 to 804 SFCs per 10^6^ T cells in H-2^b^ mice and 40 to 1178 SFCs per 10^6^ T cells in H-2^d^ mice. Interestingly, H-2^b^ and H-2^d^ have eleven shared peptides recognized by both mouse alleles. 

### 3.3. The CD8+ and CD4+ T Cell Recognition Features of the ZIKV Antigens

To further validate the immune reaction and cytokines induced by these above-positive peptides, splenocytes were stimulated with a positive peptide individually and the IFN-γ, TNF-α, and IL-2 secreting of the antigen-specific CD8+ and CD4+ T cells was detected. For H-2^b^ mice, 3 peptides presented positive for three cytokines of IFN-γ/TNF-α/IL-2 in CD8+ T cells and 13 peptides in CD4+ T cells (Figure 4). Peptides such as E_640-659_ and NS5_2955-2973_ in CD8+ T cells performed strongly, producing three cytokines. For H-2^d^ mice, six peptides presented positive for three cytokines in CD8+ T cells and three peptides in CD4+ T cells (Appendix A). Peptides such as NS1_1054-1071_ and NS4_2349-2367_ performed strongly, with three cytokines producing in CD8+ T cells. Other peptides performed immune activation with production of two or individual cytokines. Taken together, these results demonstrate a distinct CD8+ and CD4+ T cell recognition of the epitope spectrum of ZIKV.

### 3.4. The Immunodominant Hotspots of ZIKV Recognized by CD8+ T Cells in Mice

To further identify the exact short epitopes (8–11 amino acids) recognized by CD8+ T cells within the overlapping 20-mer peptides that tested positive in the screening, we predicted the potential short CD8+ T cell epitopes through the binding motif of H-2 class I molecules (D^b^, K^b^, D^d^ and K^d^). A total of 102 short epitope candidates were predicted, 45 were specific for H-2D^b^, 33 for H-2K^b^, 6 for H-2D^d^ and 18 for H-2K^d^. Through the IFN-γ-ELISPOT using the splenocytes from mice infected with ZIKV, a total of 20 H-2D^b^, 15 H-2K^b^, 2 H-2D^d^, 12 H-2K^d^ and 2 H2-I restricted epitopes were identified (Table 1 and Table 2). For H-2^b^ mice, the positive epitopes distributed among prM (3), E (9), NS1 (1), NS2 (4), NS3 (5), NS4 (7) and NS5 (6) (Figure 5A); for H-2^d^ mice, the positive epitopes distributed among prM (1), E (3), NS1 (2), NS3 (3), NS4 (5) and NS5 (2). Importantly, the distribution of the CD8+ T cell epitopes also showed hotspot characteristics, and the immunodominant regions E_294-302_ and NS4_2351-2360_ presented distinct immunodominance hierarchy in H-2^b^ and H-2^d^ mice.

To further validate the T-cell activation of ZIKV-derived CD8+ T cell epitopes from each protein in H-2^b^ mice, splenocytes were stimulated with each positive peptide to detect the frequency of IFNγ-producing CD8+ T cells. E_294–302_, E_334–355_, NS1_1237–1245_, NS2_1479–1486_, NS3_1759–1767_, NS4_2140–2147_ and NS5_2839–2848_ were the immunodominant epitopes, and induced a high frequency of IFNγ- expressing cells (Figure 5B,C). Furthermore, we synthesized specific tetramers of these immunodominant epitopes, and found that E_294–302_ and NS2_1479–1486_ tetramer-positive CD8+ T cells were detected in the splenocytes of ZIKV-infected mice (Figure 5D).

## 4. Discussion

C57BL/6 and BALB/c mice models are widely used for the pathogenesis study of ZIKV infection and vaccine development [27,28,38]. Yu, et al. compared the neurological manifestation for Zika virus infection in C57BL/6, Kunming, and the BALB/c mouse model, and found C57BL/6 owned the highest susceptibility and pathogenicity to the nervous system, while BALB/c associated with similar ocular findings to clinical cases [36]. Additionally, the strain of two mice had a different immune responses preponderance, Th1 immune response and IFN-γ production are dominant for C57BL/6, while Balb/C triggers more of the Th2 immune response and humoral response [37]. The difference in the T-cell response could be due to the fact that the MHC I locus of Balb/c mice is H-2^d^, while C57BL/6 is H-2^b^ [38]. 

Here, we developed a whole genome peptide library of ZIKV to investigate the overall antigen-specific T-cell-mediated immunity in wild-type model mice (C57BL/6 and BALB/c). Previous studies indicated that DENV (Dengue virus) dominant epitopes were within NS3, NS4B, and NS5 [39,40], whereas the major T-cell antigens of HCV (Hepatitis C virus) were located in NS3, NS4A and NS5 [41,42,43]. However, only a few studies have demonstrated T-cell epitopes of ZIKV from envelope proteins [28,38]. Our data shows that T-cell response-targeted ZIKV protein profiles in H-2^b^ and H-2^d^ mice were obviously different. Both structural and non-structural proteins appeared to be targets of the anti-ZIKV T-cell response in H-2^b^ mice, with E protein the primary target. However, non-structural proteins (NS1, NS3, NS4) showed a strong T-cell reaction in H-2^d^ mice. 

The difference in the T-cell response to immunodominant proteins (E protein) between ZIKV and other flaviviruses is very interesting. This is mainly possible due to the difference of species or alleles that we mentioned above. Additionally, there were 11/47 peptides from E protein inducing a high frequency of IFN-γ of CD8+ T cells in H-2^b^ mice, which means shorter immunodominant epitopes of E protein recognized by H-2^b^ than non-structural protein after ZIKV infection.

We provide a broad map of the T-cell response to ZIKV with identification of 91 and 39 peptides that target all viral proteins in H-2^b^ and H-2^d^ mice, respectively. The difference of MHC I locus may affect the recognition of peptides for T cells. The E, NS2, NS3 and NS5 protein induced a high frequency of IFN-γ-expressing CD8+ T cells, while E and NS4 responded to CD4+ T cell. Here we have a systematic analysis of the different activation characteristics of ZIKV proteins in CD8+ and CD4+ T cells with cytokines secreting, the NS4 protein libraries had more immunodominant peptides responding to CD4 subsets, which corresponds to the immune-thermogram analysis. These results demonstrated distinct dominance features of protein libraries to induce virus-specific CD8+/CD4+ T cells.

Moreover, multiple immunodominant epitopes such as E_294-302_ recognized by CD8+ T cells in H-2^b^ mice were highly conserved to other flaviviruses. Previous studies have found that T-cell immunity to ZIKV and DENV induced responses that are cross-reactive with other flaviviruses in both humans and HLA transgenic mice [44]. Peptides and epitopes of ZIKV we identified in C57BL/6 and BALB/c mice were important for understanding the characterization of ZIKV cross-protective immunity.

Among the positive peptides in H-2^b^ and H-2^d^ mice, respectively, the dominant epitopes of E_283-302,_ NS1_796-815_, NS4_2130-2149_, NS5_2519-2536_ and NS4_2387-2406_ were located at the junction of proteins. ZIKV, in the same way as like other flaviviruses, encodes a single polyprotein that is cleaved co-and post-translationally by cellular and viral proteases [45]. Identification of CD8+ T cell epitopes through proteasome cleavage site predictions reveals peptides that can bind to major histocompatibility complex (MHC I) molecules; the C-terminus of peptides presented by MHC I molecules result from proteasome cleavage [46,47]. It is possible that the cleavage sites of adjacent proteins are more susceptible to the protease; therefore, the processed epitopes are abundant for presentation by H-2 molecules and recognized by T cells on the surface of the flavivirus-infected cells.

## 5. Conclusions

In summary, our current study characterizes the mouse allele-dependent immune hierarchy against the whole ZIKV proteome, broaden the whole map, and draw the hotspots of the CD8+ T cell and CD4+ T cell epitope recognition profile of the virus. Our results serve to understand the T-cell immunogenic feature of ZIKV and may shed light on vaccine development.

## Figures and Tables

**Figure 1 viruses-14-02332-f001:**
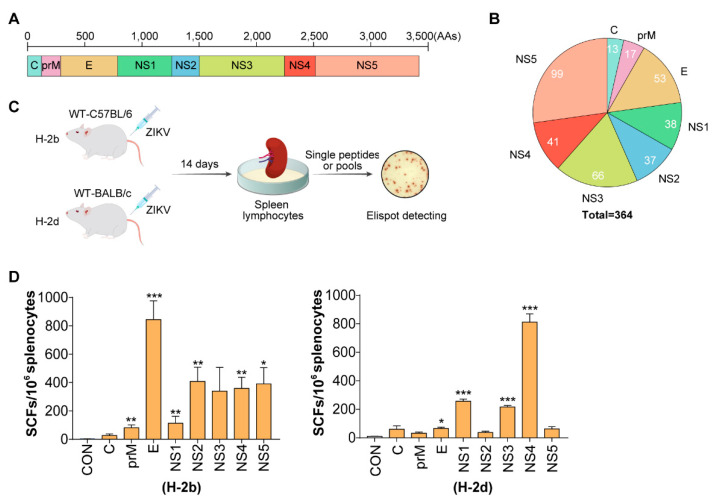
Design of overlapping ZIKV peptides and T-cell response to ZIKV proteins in H-2^b^ and H-2^d^ mice. (**A**) The full-length sequence (3423 amino acids) of the ZIKV proteome: C, prM, E, NS1, NS2, NS3, NS4, NS5. (**B**) Numbers of overlapping peptides in the eight ZIKV proteins. (**C**) Experimental flow chart: wild-type C57BL/6 and BALB/c mice (*n* = 6 per group) were infected with 10^4^ FFU of ZIKV. Mice were sacrificed at 14 days-post-infection (d.p.i.) and splenocytes isolated for ELISPOT testing. (**D**) Splenocytes were stimulated with difference protein libraries (C, prM, E, NS1, NS2, NS3, NS4, NS5) and detected with ELISPOT. A Student’s *t*-test was performed Error bars represent SEM; * *p* < 0.05; ** *p* < 0.01; *** *p* < 0.001.

**Figure 2 viruses-14-02332-f002:**
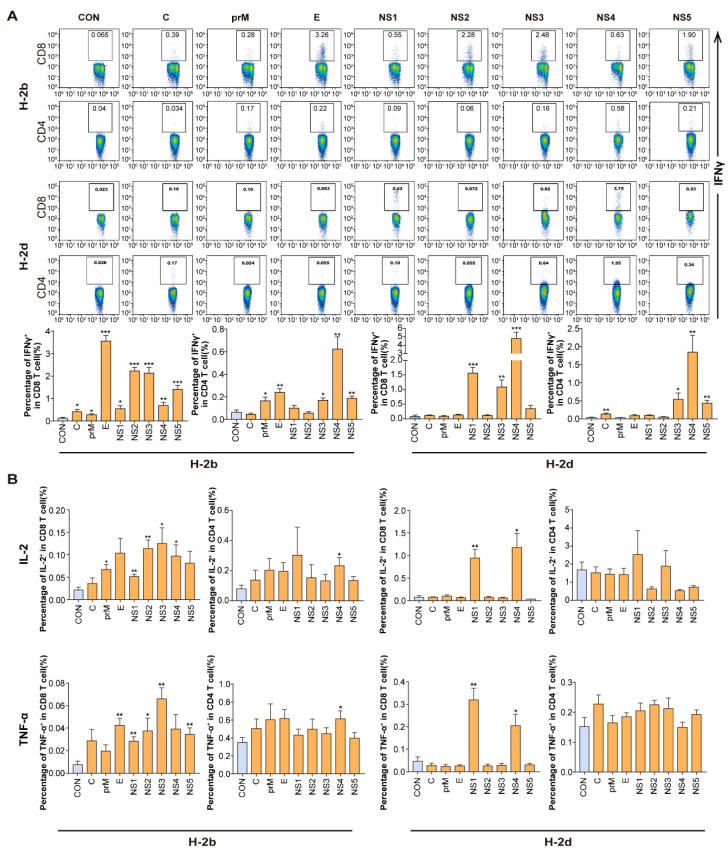
Characterization of ZIKV proteins recognized by CD8+ and CD4+ T cells. (**A**,**B**) Wild-type C57BL/6 and BALB/c mice (*n* = 6 per group) were infected with 10^4^ FFU of ZIKV, splenocytes were harvested at 14 d.p.i. and stimulated with different protein libraries (C, prM, E, NS1, NS2, NS3, NS4, NS5) to assess cytokines production of IFN-γ (**A**), TNF-α and IL-2 (**B**) by ICS in CD8+ and CD4+ T cells. A Student’s *t*-test was performed Error bars represent SEM; * *p* < 0.05; ** *p* < 0.01; *** *p* < 0.001.

**Figure 3 viruses-14-02332-f003:**
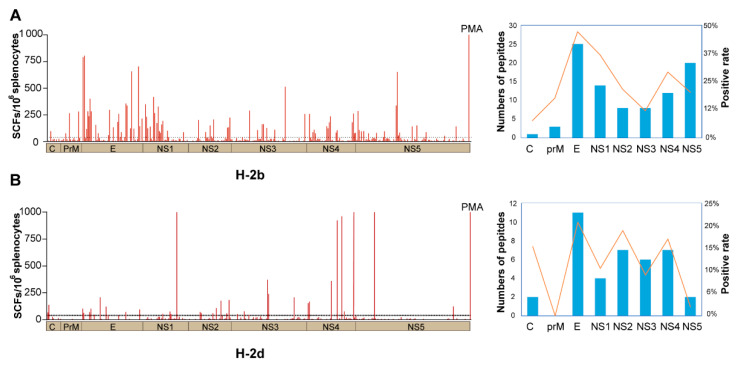
Mapped peptides according to location in the ZIKV polyprotein of H-2^b^ and H-2^d^ mice. Wild-type C57BL/6 and BALB/c mice were infected with 10^4^ FFU of ZIKV, splenocytes were harvested at 14 d.p.i. and stimulated with the indicated matrix peptide pools. A total of 364 peptides were screened by IFN-γ-ELISPOT assays, with PMA as a positive control. (**A**,**B**) Left shows SFCs of 364 peptides distributed in the ZIKV polyprotein in H-2^b^ (**A**) and H-2^d^ (**B**) mice (*n* = 3 per peptide). Right shows the number of positive peptides for each protein, SFCs ≥ 40 in H-2^b^ means positive.

**Figure 4 viruses-14-02332-f004:**
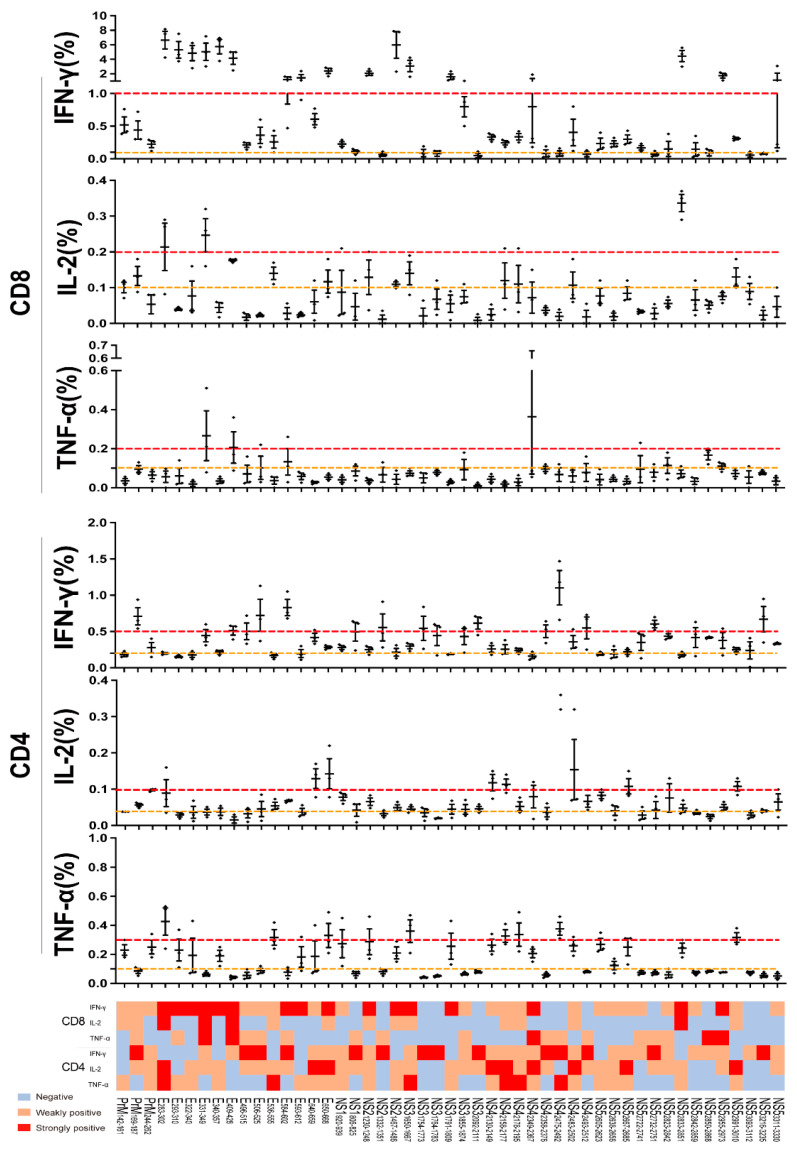
Peptides immunothermogram analysis of CD8+/CD4+ T cell in H-2^b^ and H-2^d^ mice. Wild-type C57BL/6 (H-2^b^) mice were infected with 10^4^ FFU of ZIKV, splenocytes were harvested at 14 d.p.i. and stimulated with above-positive peptides to assess cytokines production by ICS. The percentages and heat map analysis of IFN-γ, TNF-α and IL-2 produced in CD8+/CD4+T cells in H-2^b^ and H-2^d^ mice (*n* = 3 per peptide). Dashed lines between red and yellow are weakly positive, beyond red are strongly positive.

**Figure 5 viruses-14-02332-f005:**
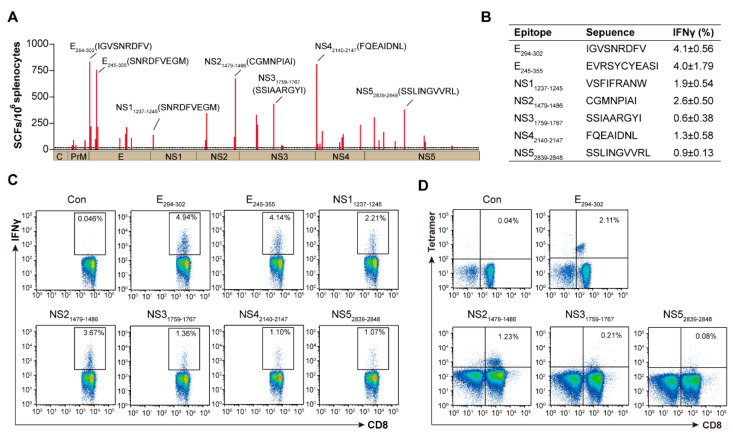
Identification of ZIKV epitopes recognized by CD8+ T cells in H-2^b^ mice. Wild-type C57BL/6 (H-2^b^) mice were infected with 10^4^ FFU of ZIKV, splenocytes were harvested at 14 d.p.i. and stimulated with epitopes (H-2^b^: Kb and Db) predicted from above-positive peptides. (**A**) Thirty-nine epitopes were identified by IFN-γ-ELISPOT assays. (**B**,**C**) The seven strongest positive epitopes from each protein are marked in the table (**B**). The percentages of IFN-γ produced in CD8+ T cells by stimulation with seven epitopes are represented, Con means without epitope stimulation (**C**). (**D**) Five tetramers were synthesized from seven positive epitopes and expression was determined by flow cytometry (*n* = 4).

**Table 1 viruses-14-02332-t001:** H-2^b^ peptides of ZIKV.

Name	Peptides Sequence	SCFs/10^6^	CD4/CD8	Epitopes Sequence	MHC	SCFs/10^6^
C_18-37_	KRGVARVSPFGGLKRLPAGL	98	NA			
PrM_142-161_	GEAISFPTTLGMNKCYIQIM	76	CD8	ISFPTTLGM	Db	43
				TLGMNKCYI	Db	94
PrM_169-187_	ATMSYECPMLDEGVEPDDV	265	CD8/CD4	MSYECPML	Db	42
PrM_244-262_	LIRVENWIFRNPGFALAAAA	281	CD8/CD4	IFRNPGFAL	Kb	90
E_283-302_	LLIAPAYSIRCIGVSNRDFV	792	CD8	IGVSNRDFV	Db	836
E_293-310_	CIGVSNRDFVEGMSGGTWV	804	CD8	SNRDFVEGM	Db	220
E_312-331_	DVVLEHGGCVTVMAQDKPTV	118	NA			
E_322-340_	TVMAQDKPTVDIELVTTTV	284	CD8			
E_331-349_	VDIELVTTTVSNMAEVRSY	238	CD8	TTVSNMAEV	Db	104
E_340-357_	VSNMAEVRSYCYEASISDMA	402	CD8	EVRSYCYEASI	Kb	758
				RSYCYEASI	Db	218
E_350-369_	CYEASISDMASDSRCPTQGE	284	NA			
E_389-408_	RGWGNGCGLFGKGSLVTCAK	156	NA			
E_409-428_	FACSKKMTGKSIQPENLEYR	78	CD8			
E_496-515_	MNNKHWLVHKEWFHDIPLPW	60	CD4			
E_506-525_	EWFHDIPLPWHAGADTGTPH	296	CD8/CD4			
E_536-555_	KDAHAKRQTVVVLGSQEGAV	132	CD8	VLGSQEGAV	Kb	110
E_575-593_	SSGHLKCRLKMDKLRLKGV	184	NA			
E_584-602_	KMDKLRLKGVSYSLCTAAF	260	CD8/CD4			
E_593-612_	VSYSLCTAAFTFTKIPAETL	42	CD8	VSYSLCTAA	Kb	153
				AAFTFTKI	Kb	214
E_603-622_	TFTKIPAETLHGTVTVEVQY	88	NA			
E_630-649_	KVPAQMAVDMQTLTPVGRLI	40	NA	MAVDMQTLTPV	Db	114
E_640-659_	QTLTPVGRLITANPVITEST	357	CD8/CD4			
E_650-668_	TANPVITESTENSKMMLEL	338	CD8			
E_679-697_	IGVGEKKITHHWHRSGSTI	124	NA			
E_688-706_	HHWHRSGSTIGKAFEATVR	658	NA			
E_705-724_	VRGAKRMAVLGDTAWDFGSV	120	NA			
E_744-763_	KSLFGGMSWFSQILIGTLLM	704	NA			
E_754-770_	SQILIGTLLMWLGLNTK	144	NA			
E_771-788_	NGSISLMCLALGGVLIFL	216	NA			
NS1_796-815_	VGCSVDFSKKETRCGTGVFV	350	NA			
NS1_806-825_	ETRCGTGVFVYNDVEAWRDR	232	CD8/CD4	TGVFVYNDV	Kb	144
NS1_816-835_	YNDVEAWRDRYKYHPDSPRR	124	NA			
NS1_835-854_	RLAAAVKQAWEDGICGISSV	140	NA			
NS1_864-883_	SVEGELNAILEENGVQLTVV	418	NA			
NS1_835-855_	EENGVQLTVVVGSVKNPMWR	266	NA			
NS1_874-893_	RGPQRLPVPVNELPHGWKAW	174	NA			
NS1_903-922_	NELPHGWKAWGKSYFVRAAK	328	NA			
NS1_913-929_	GKSYFVRAAKTNNSFVV	98	NA			
NS1_920-939_	AAKTNNSFVVDGDTLKECPL	82	NA			
NS1_930-949_	DGDTLKECPLKHRAWNSFLV	160	NA			
NS1_940-958_	KHRAWNSFLVEDHGFGVFH	194	NA			
NS1_976-995_	AVIGTAVKGKEAVHSDLGYW	102	NA			
NS1_1106-1125_	CCRECTMPPLSFRAKDGCWY	88	NA			
NS2_1230-1248_	KVRPALLVSFIFRANWTPR	202	CD8	VSFIFRAN	Kb	92
				VSFIFRANW	Kb	348
NS2_1283-1320_	LAIRAMVVPRTDNITLAILA	90	NA			
NS2_1322-1341_	TCGGFMLLSLKGKGSVKKNL	152	NA			
NS2_1332-1351_	KGKGSVKKNLPFVMALGLTA	48	CD4			
NS2_1352-1371_	VRLVDPINVVGLLLLTRSGK	208	NA			
NS2_1467-1486_	REIILKVVLMTICGMNPIAI	130	CD8	VLMTICGM	Db	123
				CGMNPIAI	Db	676
NS2_1477-1496_	TICGMNPIAIPFAAGAWYVY	134	NA			
NS2_1487-1506_	PFAAGAWYVYVKTGKRSGAL	224	NA			
NS3_1650-1667_	GLYGNGVVIKNGSYVSAI	291	CD8	VVIKNGSYV	Db	332
				NGSYVSAI	Db	236
NS3_1716-1734_	KTRLRTVILAPTRVVAAEM	108	NA			
NS3_1754-1773_	HSGTEIVDLMCHATFTSRLL	162	CD4			
NS3_1764-1783_	CHATFTSRLLQPIRVPNYNL	164	CD4			
NS3_1791-1809_	FTDPSSIAARGYISTRVEM	128	CD8	SSIAARGYI	Db	436
NS3_1855-1874_	TDHSGKTVWFVPSVRNGNEI	111	CD8/CD4	PSVRNGNEI	Kb	46
				SVRNGNEI	Db	39
NS3_1936-1955_	ILDGERVILAGPMPVTHASA	514	NA			
NS3_2092-2111_	LKPRWMDARVCSDHAALKSF	259	CD4			
NS4_2130-2149_	GTLPGHMTERFQEAIDNLAV	259	CD8/CD4	FQEAIDNL	Db	812
				FQEAIDNLAV	Db	56
NS4_2158-2177_	RPYKAAAAQLPETLETIMLL	88	CD8/CD4	QLPETLETI	Db	58
NS4_2168-2187_	PETLETIMLLGLLGTVSLGI	112	NA			
NS4_2178-2194_	GLLGTVSLGIFFVLMRNKGI	76	CD8/CD4	VSLGIFFVLM	Kb	178
NS4_2275-2293_	LERTKSDLSHLMGRREEGA	144	NA			
NS4_2284-2303_	HLMGRREEGATIGFSMDIDL	124	NA			
NS4_2092-2116_	TIGFSMDIDLRPASAWAIYA	180	NA			
NS4_2294-2313_	RPASAWAIYAALTTFITPAV	236	NA			
NS4_2349-2367_	MGKGMPFYAWDFGVPLLMI	84	CD8	YAWDFGVPL	Kb	120
				YAWDFGVPLL	Kb	150
NS4_2358-2376_	WDFGVPLLMIGCYSQLTPL	106	CD4			
NS4_2475-2492_	LWEGSPNKYWNSSTATSL	180	CD4			
NS4_2483-2502_	YWNSSTATSLCNIFRGSYLA	263	CD8/CD4	CNIFRGSYL	Kb	236
NS4_2493-2512_	CNIFRGSYLAGASLIYTVTR	77	CD4			
NS5_2503-2520_	GASLIYTVTRNAGLVKRR	82	NA			
NS5_2519-2536_	RRGGGTGETLGEKWKARL	286	NA			
NS5_2527-2545_	TLGEKWKARLNQMSALEFY	108	NA			
NS5_2546-2525_	SYKKSGITEVCREEARRALK	96	NA			
NS5_2566-2585_	DGVATGGHAVSRGSAKLRWL	98	NA			
NS5_2605-2623_	GGWSYYAATIRKVQEVKGY	76	CD8	WSYYAATI	Kb	308
NS5_2667-2685_	IGESSSSPEVEEARTLRVL	81	CD8/CD4	EVEEARTL	Db	168
NS5_2722-2741_	YGGGLVRVPLSRNSTHEMYW	40	CD8			
NS5_2732-2751_	SRNSTHEMYWVSGAKSNTIK	96	CD8/CD4			
NS5_2823-2842_	TWAYHGSYEAPTQGSASSLI	338	CD8/CD4			
NS5_2833-2851_	PTQGSASSLINGVVRLLSK	654	CD8	SSLINGVVRL	Db	382
NS5_2842-2859_	INGVVRLLSKPWDVVTGV	58	CD8/CD4			
NS5_2850-2868_	SKPWDVVTGVTGIAMTDTT	84	CD4			
NS5_2955-2973_	LVDKEREHHLRGECQSCVY	142	CD8			
NS5_2991-3010_	GSRAIWYMWLGARFLEFEAL	152	CD8	RAIWYMWL	Kb	
				GSRAIWYM	Db	
NS5_3064-3083_	SRFDLENEALITNQMEKGHR	88	NA			
NS5_3093-3112_	TYQNKVVKVLRPAEKGKTVM	88	CD4			
NS5_3216-3235_	WKPSTGWDNWEEVPFCSHHF	54	CD4	TGWDNWEEV	Db	40
NS5_3311-3330_	PTGRTTWSIHGKGEWMTTED	142	CD8/CD4			

**Table 2 viruses-14-02332-t002:** H-2^d^ epitopes of ZIKV.

Name	Peptides Sequence	SCFs/10^6^	CD4/CD8	Epitopes Sequence	MHC	SCFs/10^6^
C_1-19_	MKNPKKKSGGFRIVNMLKR	68	CD4			
C_10-27_	GFRIVNMLKRGVARVSPF	138	NA			
E_283-302_	LLIAPAYSIRCIGVSNRDFV	104	NA	AYSIRCIGV	Kd	124
E_293-311_	CIGVSNRDFVEGMSGGTWV	64	NA			
E_340-359_	VSNMAEVRSYCYEASISDMA	72	CD4	SYCYEASI	Kd	524
				CYEASISDM	Kd	108
E_350-369_	CYEASISDMASDSRCPTQGE	104	CD8			
E_380-398_	YVCKRTLVDRGWGNGCGLF	46	NA			
E_429-448_	IMLSVHGSQHSGMIVNDTGH	208	CD4/CD8			
E_477-496_	GLDCEPRTGLDFSDLYYLTM	122	NA			
E_496-515_	MNNKHWLVHKEWFHDIPLPW	44	NA			
E_584-600_	KMDKLRLKGVSYSLCTAAF	40	NA	SYSLCTAA	Kd	84
E_640-659_	QTLTPVGRLITANPVITEST	72	CD4			
E_754-770_	SQILIGTLLMWLGLNTK	96	CD4			
NS1_940-958_	KHRAWNSFLVEDHGFGVFH	56	CD4/CD8			
NS1_996-1015_	IESEKNDTWRLKRAHLIEMK	76	CD4/CD8			
NS1_1006-1025_	LKRAHLIEMKTCEWPKSHTL	52	CD4			
NS1_1054-1071_	YRTQMKGPWHSEELEIRF	1000	CD8	KGPWHSEEL	Dd	376
				GYRTQMKGPW	Kd	174
NS2_1239-1256_	FIFRANWTPRESMLLALA	72	CD4			
NS2_1247-1266_	PRESMLLALASCLLQTAISA	64	CD8			
NS2_1342-1360_	PFVMALGLTAVRLVDPINVV	60	CD4/CD8			
NS2_1371-1390_	KRSWPPSEVLTAVGLICALA	108	CD4			
NS2_1410-1428_	LIVSYVVSGKSVDMYIERA	176	CD4			
NS2_1457-1476_	SLVEDDGPPMREIILKVVLM	60	CD4			
NS2_1477-1496_	TICGMNPIAIPFAAGAWYVY	184	CD4			
NS3_1544-1561_	QEGVFHTMWHVTKGSALR	52	CD4			
NS3_1602-1621_	VPPGERARNIQTLPGIFKTK	79	NA			
NS3_1640-1659_	PILDKCGRVIGLYGNGVVIK	49	NA	LYGNGVVI	Kd	80
NS3_1791-1809_	FTDPSSIAARGYISTRVEM	372	CD4/CD8	GYISTRVEM	Kd	174
NS3_1800-1819_	RGYISTRVEMGEAAAIFMTA	240	CD8			
NS3_2002-2020_	QDGLIASLYRPEADKVAAI	208	CD8	LYRPEADKV	Kd	158
NS4_2112-2129_	KEFAAGKRGAAFGVMEAL	153	NA			
NS4_2120-2139_	GAAFGVMEALGTLPGHMTER	168	CD4/CD8			
NS4_2304-2323_	RPASAWAIYAALTTFITPAV	360	CD4/CD8	IYAALTTFI	Kd	128
NS4_2349-2367_	MGKGMPFYAWDFGVPLLMI	924	CD8	KGMPFYAWDF	Dd	564
				FYAWDFGVPLL	Kd	230
NS4_2387-2406_	AHYMYLIPGLQAAAARAAQK	963	CD4/CD8	LIPGLQAAAARAAQK	H2-I	168
NS4_2405-2423_	QKRTAAGIMKNPVVDGIVV	78	NA			
NS4_2475-2492_	LWEGSPNKYWNSSTATSL	1178	CD4/CD8	GSPNKYWNSSTATSL	H2-I	534
NS5_2638-2657_	SYGWNIVRLKSGVDVFHMAA	1000	CD4/CD8	SYGWNIVRL	Kd	66
NS5_3272-3291_	ETACLAKSYAQMWQLLYFHR	124	CD4/CD8	SYAQMWQLL	Kd	96

## Data Availability

All data required to interpret the data are provided in the main document or the Appendix A. Further data are available from the corresponding author upon reasonable request.

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
