# Peer review of "The CD8+ and CD4+ T Cell Immunogen Atlas of Zika Virus Reveals E, NS1 and NS4 Proteins as the Vaccine Targets"

_viruses, 2022, doi:10.3390/v14112332_

Round 1

Reviewer 1 Report

T cell response plays pivotal role in clearance of virus infection,thus screening and identifying the T cell epitopes and T cell immunogens derived from virus proteome is a fundamental work. This study used mice infection models and overlapping peptides spanning the whole proteome of Zika virus along with ELISpot, ICS and tetramer staining to map the profile of CD4+ T cell and CD8+ T cell immunogens and epitopes. The methods are appropriate and classical and the results are sounded. A series of predominant immunogenic peptides and T cell epitopes of Zika proteins are verified. This will benefit for the investigation of cellular immunity of Zika infection and the design of vaccines inducing T cell response. Several minor comments are listed below:

1. In Figure 1D and Figure 3A/B, the longitudinal title “SCF/106 T cells” seems to should be “SFCs/106 splenocytes”.

2. In ELISpot and ICS assays, whether DMSO with the mean concentration in peptide/splenocytes co-incubation well was added into the control well (spelnocytes alone) should be described.

3. How to prevent the binding between acid peptide and alkaline peptide when prepare the peptides pool?

4. when SFCs = 38 in the control well, dose SFCs = 41 in the peptide/splenoctyes con-incubation well still mean positive response?

5. How to explain only two tetramers showed positive response in the splenocytes staining assay?

Reviewer 2 Report

Zhang and colleagues presented very elegant and convincing data on a differential immunogenicity profile of CD8+ and CD4+ T cells for ZIKV proteins in C57BL/6 (H-2b) and BALB/c (H-2d) mice. The cellular immunity antigens were variants among the different murine alleles. I consider this study an additional report of knowledge about mouse models of ZIKV. In my view, the authors should reevaluate a few points, among them:

1)      The authors should biologically clarify the choice to use two mouse models, emphasizing the immunological profile of each strain. In this context, the authors could discuss the differences (or not) in the responses of T cell subpopulations to the peptides in each of the strains.

2)      Both animal models have been widely used in vaccine models and Zika immunopathogenesis studies. The authors discussed very little about these two aspects.

Some bibliographies that should be added:

Yu J, Liu X, Ke C, Wu Q, Lu W, Qin Z, He X, Liu Y, Deng J, Xu S, Li Y, Zhu L, Wan C, Zhang Q, Xiao W, Xie Q, Zhang B, Zhao W. Effective Suckling C57BL/6, Kunming, and BALB/c Mouse Models with Remarkable Neurological Manifestation for Zika Virus Infection. Viruses. 2017 Jun 29;9(7):165. doi: 10.3390/v9070165.

Manangeeswaran M, Ireland DD, Verthelyi D. Zika (PRVABC59) Infection Is Associated with T cell Infiltration and Neurodegeneration in CNS of Immunocompetent Neonatal C57Bl/6 Mice. PLoS Pathog. 2016 Nov 17;12(11):e1006004. doi: 10.1371/journal.ppat.1006004. 

Wang R, Liao X, Fan D, Wang L, Song J, Feng K, Li M, Wang P, Chen H, An J. Maternal immunization with a DNA vaccine candidate elicits specific passive protection against post-natal Zika virus infection in immunocompetent BALB/c mice. Vaccine. 2018 Jun 7;36(24):3522-3532. doi: 10.1016/j.vaccine.2018.04.051.

Medina-Magües LG, Gergen J, Jasny E, Petsch B, Lopera-Madrid J, Medina-Magües ES, Salas-Quinchucua C, Osorio JE. mRNA Vaccine Protects against Zika Virus. Vaccines (Basel). 2021 Dec 10;9(12):1464. doi: 10.3390/vaccines9121464.
